# An Arithmetic-Trigonometric Optimization Algorithm with Application for Control of Real-Time Pressure Process Plant

**DOI:** 10.3390/s22020617

**Published:** 2022-01-13

**Authors:** P. Arun Mozhi Devan, Fawnizu Azmadi Hussin, Rosdiazli B. Ibrahim, Kishore Bingi, M. Nagarajapandian, Maher Assaad

**Affiliations:** 1Department of Electrical and Electronics Engineering, Universiti Teknologi PETRONAS, Seri Iskandar 32610, Malaysia; arundevaeie@gmail.com (P.A.M.D.); rosdiazli@utp.edu.my (R.B.I.); 2School of Electrical Engineering, Vellore Institute of Technology, Vellore 632014, Tamil Nadu, India; kishore.bingi@vit.ac.in; 3Department of Electronics and Instrumentation Engineering, Sri Ramakrishna Engineering College, Coimbatore 641022, Tamil Nadu, India; nagarajapandian.m@srec.ac.in; 4Department of Electrical and Computer Engineering, Ajman University, Ajman 666688, United Arab Emirates; m.assaad@ajman.ac.ae

**Keywords:** arithmetic–trigonometric optimization, benchmark functions, dead-time processes, fractional-order controller, PID control, process control, trigonometric functions

## Abstract

This paper proposes a novel hybrid arithmetic–trigonometric optimization algorithm (ATOA) using different trigonometric functions for complex and continuously evolving real-time problems. The proposed algorithm adopts different trigonometric functions, namely sin, cos, and tan, with the conventional sine cosine algorithm (SCA) and arithmetic optimization algorithm (AOA) to improve the convergence rate and optimal search area in the exploration and exploitation phases. The proposed algorithm is simulated with 33 distinct optimization test problems consisting of multiple dimensions to showcase the effectiveness of ATOA. Furthermore, the different variants of the ATOA optimization technique are used to obtain the controller parameters for the real-time pressure process plant to investigate its performance. The obtained results have shown a remarkable performance improvement compared with the existing algorithms.

## 1. Introduction

In the era of advanced technological development, the complexity of implementing real-time processes and applications requires effective metaheuristic optimization tuning techniques to meet the global demands with more reliability [1]. The traditional optimization techniques determine the local minima based on the analytical calculations using the relevant processes models, which generally produce a single optimal solution at each run [2,3]. On the other hand, the metaheuristic techniques overcome the conventional methods due to the effective gradient-free mechanisms and the excellent handling of local optima of the respective functions [4,5]. In most metaheuristic algorithms, the optimal solution is found by increasing or decreasing the desired objective function [6,7]. Every optimization algorithm has its challenges in facing the real-time applications that are diverse in nature, multimodal system structure, complexity in implementation, bounded space limitations, longer processing time, flexibility in parameter modifications, and application-specific requirements [8,9,10].

In general, there are different categories of metaheuristic optimization algorithms available. The categorization is based on inspiration: nature-based, swarm and evolutionary intelligence, physics-based, and human activities-based techniques [4,11]. Most metaheuristic optimization techniques identify the optimum solution area by incorporating the two critical search phases: exploration and exploitation [12]. The exploration stage performs the function of finding the desired local optimal solution in the global search area. On the other hand, the exploitation phase will serve the different possible solutions in the desired search area [13]. This process improves the effective search area selection and local optima improvement. These two search phases must be in an apt balance to ensure the efficient performance of the optimization [14]. In the early optimization techniques, only a single optimal solution is possible for each algorithm run. Later developments are motivated by the natural phenomena to solve the multi-objective processes using three realistic procedures: select, recombine, and mutate [15,16].

Among the different optimization techniques, the SCA forms the base research idea for this article [17]. Compared with the other metaheuristic techniques, SCA has a simple structure and is flexible towards selecting the parameters concerning the application needs. The SCA uses the analytical model of the trigonometric functions (i.e., sin and cos) to find the initial arbitrary search to identify the best solutions. Population and local searches are the two critical strategies adopted by the SCA to yield the best solution possible in both search phases (exploration and exploitation). The SCA faces the problem of premature convergence in both stages, and this issue is solved by combining the chaotic-based search area mechanism and the cultural algorithm technique termed chaotic cultural SCA [18]. Additionally, some hybrid optimization techniques were developed by combining SCA with other algorithms to improve the local search problems. Researchers developed such an improved algorithm by combining the flower pollination algorithm [19]. Here, the SCA performs the internal search operation to solve the algorithm’s local optimal search area problem. Similarly, the authors in [20] combined SCA and the genetic algorithm, resulting in a new hybrid steady-state genetic algorithm to solve the early convergence issues encountered by the many engineering applications. Tawhid and Savsani proposed the optimization for multiple objective functions by using the crowd-distance method to obtain the different optimal search areas for retaining the presence of diversity among those optimal solutions [21].

Abualigah et al. in [1] proposed the AOA by using the arithmetic operators such as multiplication, division, addition, and subtraction to solve the single-objective problems. Later, Premkumar et al. improvised this technique for the multiple-objective applications and termed it real-world constrained multi-objective optimization problems [22]. An improved version of the AOA is proposed by Panga et al. in [23] by applying the square and cubic functions to the bounded values in the exploration and exploitation phases. They obtained performance improvement only in the lower dimension functions whose global minima values are close to zero. Zheng et al., in [24], proposed another modification for the AOA technique by forcefully switching the optimization function based on a new random probability parameter. All the above techniques improved the ability of AOA to reach the best position; however, the search agents cannot relocate their position to identify the best local minima due to their small step-changing capability. In recent years, the application of SCA and AOA techniques has extended to various engineering fields such as image processing [25], engineering design problems [26,27], feature selection [28,29], fault diagnosis [30], text categorization [31], power systems [32,33], parameter estimation [34], cruise control [35], scheduling applications [36], object tracking [37], energy production [38], controller parameters optimization [39], and benchmark functions [40].

The above literature studies conclude that using higher population groups to find an optimal solution does not assure the desired optimal value because of the increased number of random optimal solutions and continuously increasing real-world problems [41,42]. Nevertheless, the search mechanism is common for all metaheuristic algorithms, i.e., exploration and exploitation [43]. The No Free Lunch theorem also states that no such metaheuristic optimization algorithm is available to solve all types of optimization problems in simulation and real-time environments [44]. Hence, there is always an opportunity for improving the design of the existing optimization algorithms. Thus, using the SCA’s faster local search mechanism and best position identification ability of the AOA, a new hybridized arithmetic-trigonometric optimization algorithm (ATOA) is designed to faster attain the convergence rate and achieve better optimal zone finding in the exploration and exploitation phases.

Some of the significant contributions in this research article can be listed as follows:1.The proposed ATOA technique avoids premature convergence and accelerates the search mechanism using the trigonometric functions (i.e., sin, cos, and tan).2.The different combinations of the proposed trigonometric function quickly relocate its position from one local minimum to another without getting stuck and reducing the computational complexity.3.The proposed optimization is simulated and validated with 33 different benchmark functions to determine its rate of convergence and optimal solution zone identification performance.4.The ATOA optimized controller is implemented on the real-time pressure process plant to validate the proposed optimization technique.

The paper’s organization is given as follows: A comprehensive design procedure of the proposed ATOA technique is discussed in Section 2. The performance analysis of the proposed ATOA and conventional algorithms on different benchmark functions is given in Section 3. The experimental results of ATOA optimized controller parameters on the real-time pressure process plant are compared to the conventional techniques in Section 4. Summary, concluding remarks, and future directions of the proposed work are provided in Section 5.

## 2. The Arithmetic–Trigonometric Optimization Algorithm

This section gives the development of the proposed ATOA based on the inspirations from the SCA and AOA algorithms. Selecting these two specific algorithms is due to their flexibility in modifying the design structure and parameters based on the system requirements. In addition, their simple and easy implementable structure allows it to be used for different engineering problems.

### 2.1. Sine Cosine Algorithm

The SCA algorithm is also a population-based metaheuristic optimization technique. The optimization will be initiated using a group of populations with random solutions. This random set needs a continuous evaluation to reach the optimal solution based on the desired objective function. The probability of finding the best optimal solution in the single run of this metaheuristic algorithm is very low. Still, the chances will be increased if there are a sufficient number of iterations and random set solutions available. The SCA also consists of the exploration and exploitation phases to reach the global minima by locating the optimal search regions. The position update equation during this exploration phase is given as [17]
(1)$$X_{i}^{t+1}=X_{i}^{t}+r_{1}\;\times \;\sin\left(r_{2}\right)\;\times \;\left|r_{3}P_{i}^{t}-X_{i}^{t}\right|.$$

Once the area is located by exploration, the fluctuations that occur in the random solutions will be reduced in the exploitation phase. The position update equation during this exploitation phase is given as Equation (Equation 2):(2)$$X_{i}^{t+1}=X_{i}^{t}+r_{1}\;\times \;\cos\left(r_{2}\right)\;\times \;\left|r_{3}P_{i}^{t}-X_{i}^{t}\right|.$$

In Equations (Equation 1) and (Equation 2),

$X_{i}^{t}$ and $X_{i}^{t+1}$ are the positions of $i^{th}$ solution at $t^{th}$ and ${\left(t+1\right)}^{th}$ iterations.$P_{i}^{t}$ is the point of destination of $i^{th}$ solution at $t^{th}$ iteration.$r_{1}$, $r_{2}$, $r_{3}$, and $r_{4}$ are the random variables.

The parameter $r_{1}$ guides the direction of movement. This direction is the next position towards the region between the solution and the destination. The guiding principle of $r_{1}$ is defined as
(3)$$r_{1}=2-t\left(\frac{2}{T}\right),$$
where *t* and *T* are the number of current and maximum iterations, respectively.

The parameter $r_{2}$ determines how far the movement of the particles has to go either towards or away from the destination. Similarly, the parameter $r_{3}$ specifies the goal to define the distance movement using the random weights to emphasize or de-emphasize them. The equations for generating these $r_{2}$ and $r_{3}$ values are given as
(4)$$\begin{array}{cc} \; & r_{2}=2\pi \cdot rand\left(\right),\\ \; & r_{3}=2\cdot rand\left(\right). \end{array}$$

Furthermore, the choice of emphasizing or de-emphasizing effect based on the value of $r_{3}$ is as follows:(5)$$\mathrm{Effect}=\left\{\begin{array}{cc} \mathrm{Emphasize}, & \mathrm{if}\;r_{3}>1,\\ \mathrm{De}-\mathrm{emphasize}, & \mathrm{if}\;r_{3}\leq 1. \end{array}\right.$$

Finally, the parameter $r_{4}$ helps switch between the sin and cos functions during the exploration and exploitation. The range of $r_{4}$ lies in [0,1], and this switching will be performed as follows:(6)$$X_{i}^{t+1}=\left\{\begin{array}{cc} X_{i}^{t}+r_{1}\;\times \;\sin\left(r_{2}\right)\;\times \;\left|r_{3}P_{i}^{t}-X_{i}^{t}\right|, & \mathrm{if}\;r_{4}<0.5,\\ X_{i}^{t}+r_{1}\;\times \;\cos\left(r_{2}\right)\;\times \;\left|r_{3}P_{i}^{t}-X_{i}^{t}\right|, & \mathrm{if}\;r_{4}\geq 0.5. \end{array}\right.$$

The pictorial representation of switching conditions of the SCA is given in Figure 1. Here, the random variable $r_{1}$ movement towards the destination is based on the current solution and position.

### 2.2. Arithmetic Optimization Algorithm

The design of the AOA starts with the usage of four arithmetic operators as per the order given as follows: division (÷), multiplication (×), subtraction (−), and addition (+), as shown in Figure 2. The conditions for selecting the exploration and exploitation phases of the algorithm are depicted in Figure 2a. The hierarchical order of these four mathematical operators is illustrated in Figure 2b. In contrast, the mechanism of searching for the optimal solution area is shown in Figure 2c [1]. In AOA, three phases are involved in finding the best optimal solution, and they are given as follows: initialization, exploration, and exploitation. Comparing Figure 1 with Figure 2 clearly shows that the AOA follows the footsteps of the SCA in finding the optimal solution.

#### 2.2.1. Initialization

The optimization commences with the randomly generated initial candidate solutions *X*, which are obtained using Equation (Equation 7) with the order of n×d. Here, *n* is the number of solutions, and *d* is the number of variables. The best variable obtained from each iteration of the matrix *X* is considered the best optimal solution, which is given as
(7)$$X=\left[\begin{array}{ccccc} x_{1,1} & \cdots & x_{1,j} & x_{1,n-1} & x_{1,n}\\ x_{2,1} & \cdots & x_{2,j} & \cdots & x_{2,n}\\ \ldots & \cdots & \cdots & \cdots & \cdots \\ \vdots & \vdots & \vdots & \vdots & \vdots \\ x_{N-1,1} & \cdots & x_{N-1,j} & \cdots & x_{N-1,n}\\ x_{N,1} & \cdots & x_{N,j} & x_{N,n-1} & x_{N,n} \end{array}\right].$$

The math optimizer accelerated (MOA) function determines the next search phase (i.e., exploration or exploitation) of the algorithm with the help of division and multiplication operators. The equation to find the MOA coefficient is given as
(8)$$\mathrm{MOA}\left(t\right)=\min+t\left(\frac{\max-\min}{T}\right),$$
where “min” is the minimum value of MOA, “max” is the maximum value of MOA, *t* is the current iteration, and *T* is the maximum number of iterations.

The next operation in the AOA is categorized into two phases based on the MOA function value. The phase search operation is used to commence the next phase of AOA, which is decided based on the MOA value using the equation as follows:(9)$$\mathrm{Search}\;\mathrm{operation}=\left\{\begin{array}{cc} \mathrm{Exploration}, & \mathrm{if}\;r_{1}\leq \mathrm{MOA},\\ \mathrm{Exploitation}, & \mathrm{if}\;r_{1}>\mathrm{MOA}. \end{array}\right.$$

#### 2.2.2. Exploration

In the exploration operation, division and multiplication operators produce practical search operation values or results, which lead to the identification of near-optimal values for the next phase (exploitation). Here, the exploration is started based on the condition given by MOA in (Equation 9). The first operator, ÷, will initiate the searching performance task if the value of $r_{2}\leq 0.5$. The adjoining operator × will be ignored during this process until the current job is completed. Otherwise, the second operator × will start the first task if $r_{2}>0.5$ instead of the ÷ operator. The position update of the exploration phase is obtained using the equation, as given as follows.

In the exploration operation, division and multiplication operators produce effective search operation values or results, which lead to the identification of near-optimal values for the next phase (exploitation). Here, the exploration is started based on the condition given by MOA ($r_{1}\leq \mathrm{MOA}$). The first operator, ÷, will initiate the searching performance task at the beginning if $r_{2}\leq 0.5$. In the meantime, the adjoining operator, ×, will be ignored during this process until the current job is completed. Otherwise, the second operator, ×, will start the first task ($r_{2}>0.5$) instead of the ÷ operator. The position update of the exploration phase is obtained using the following equation:(10)$$x_{i,j}\left(t+1\right)=\left\{\begin{array}{cc} best\left(x_{j}\right)÷\left(\mathrm{MOP}+\epsilon \right)\;\times \;\left(\left({UB}_{j}-{LB}_{j}\right)\;\times \;\mu +\left({LB}_{j}\right)\right), & r_{2}\leq 0.5,\\ best\left(x_{j}\right)\;\times \;\mathrm{MOP}\;\times \;\left(\left({UB}_{j}-{LB}_{j}\right)\;\times \;\mu +\left({LB}_{j}\right)\right), & r_{2}>0.5. \end{array}\right.$$
where MOP is the math optimizer probability coefficient, which is calculated using
(11)$$\mathrm{MOP}\left(t\right)=1-{\left(\frac{t}{T}\right)}^{1/\alpha }$$

In Equation (Equation 11),

α is a sensitive parameter that defines the exploitation accuracy.*t* and *T* are the current and maximum number of iterations, respectively.

#### 2.2.3. Exploitation

The operators used in exploitation are subtraction and addition, which have a low dispersion rate, leading to a faster and more accessible approach to reach the target value, which yields a lower number of iterations in the optimization process of the exploration phase. Here, these exploitation operators have enhanced communication which allows better coordination between them. The exploitation search phase is conditioned with the help of the MOA value, as given in Equation (Equation 9).

In this stage, the subtraction operator will start performing the searching task if $r_{3}\leq 0.5$. The addition operator will be ignored during this searching period until the current job is completed. Otherwise, the addition operator will start the task instead if $r_{3}>0.5$ (see Figure 2c). Furthermore, exploitation search operators will search deeper in the denser regions to find a better optimal solution. It is worth mentioning that, due to the low dispersion rate of the addition and subtraction operators, the probability of getting stuck in the same search region is very minimal. If it happens, they will recover faster than other operators from that local search region. The position update in the exploitation phase is obtained using the following equation:(12)$$x_{i,j}\left(t+1\right)=\left\{\begin{array}{cc} best\left(x_{j}\right)\;-\;\mathrm{MOP}\;\times \;\left(\left({UB}_{j}-{\mathrm{LB}}_{j}\right)\;\times \;\mu +\left({LB}_{j}\right)\right), & r_{3}\leq 0.5,\\ best\left(x_{j}\right)\;+\;\mathrm{MOP}\;\times \;\left(\left(UB_{j}-LB_{j}\right)\;\times \;\mu +\left(LB_{j}\right)\right), & r_{3}>0.5. \end{array}\right.$$

### 2.3. Arithmetic–Trigonometric Optimization Algorithm

The above two metaheuristic techniques inspired us to hybridize them to create this new ATOA technique. Unlike SCA, the proposed ATOA uses four arithmetic operators instead of the random variables. The ATOA also maintains the same structure and position update mechanism as the existing AOA. On top of that, the proposed method uses trigonometric functions (i.e., sin, cos, and tan) to reposition the solution around the desired solution. This condition ensures the exploration and exploitation search spaces remain around the desired destination area. By combining all these techniques, the design of the proposed ATOA algorithm is explained underneath.

The cyclic and infinite vertical patterns of sin, cos, and tan functions allow the discovered solution to readjust around the nearby solution. This approach guarantees the desired space needs during the exploitation phase in between the two adjacent solutions. During the exploration phase, by adjusting the range of the trigonometric functions, the solutions yield the capability to search outside the space between their corresponding optimum solution zone. A position update is achieved by changing ranges of the sin, cos, and tan functions. Here, the solution repositions inside or outside their search space between itself and the adjacent key. The position update, either inside or outside, is determined by the random number $r_{2}$ used in the ATOA. Hence, a balance between switching exploration and exploitation is essential, and it is regulated based on MOA. In addition, the range of the trigonometric function is controlled by using MOP. The diverse patterns of the trigonometric functions in the range of [0,3π] are illustrated in Figure 3.

The random numbers $r_{1}$ and $r_{2}$ determine the algorithm phase and operators switching based on its value. During the standalone implementation of sin, cos, and tan functions in the proposed ATOA technique, only one trigonometric function is used in all the phases (i.e., exploration and exploitation) in between the bounded values. Thus, the position update of the standalone sin function in the exploration and exploitation phases of ATOA is obtained using the following Equations (Equation 13) and (Equation 14), respectively. Similarly, for the following trigonometric function, i.e., cos, the existing sin is replaced with cos during both phases, and the same procedure is repeated for tan.
(13)$$x_{i,j}\left(t+1\right)=\left\{\begin{array}{cc} best\left(x_{j}\right)\;÷\;\left(\mathrm{MOP}+\epsilon \right)\;\times \;\left(\sin\left({UB}_{j}-{LB}_{j}\right)\;\times \;\mu +\sin\left({LB}_{j}\right)\right), & r_{2}\leq 0.5,\\ best\left(x_{j}\right)\;\times \;\mathrm{MOP}\;\times \;\left(\sin\left({UB}_{j}-{LB}_{j}\right)\;\times \;\mu +\sin\left({LB}_{j}\right)\right), & r_{2}>0.5, \end{array}\right.$$
(14)$$x_{i,j}\left(t+1\right)=\left\{\begin{array}{cc} best\left(x_{j}\right)\;-\;\mathrm{MOP}\;\times \;\left(\sin\left({UB}_{j}-{\mathrm{LB}}_{j}\right)\;\times \;\mu +\sin\left({LB}_{j}\right)\right), & r_{3}\leq 0.5,\\ best\left(x_{j}\right)\;+\;\mathrm{MOP}\;\times \;\left(\sin\left(UB_{j}-LB_{j}\right)\;\times \;\mu +\sin\left(LB_{j}\right)\right), & r_{3}>0.5. \end{array}\right.$$

The different combinations of the trigonometric functions used during the exploration and exploitation phase are determined based on the MOA coefficient. The motivation to select the various trigonometric functions combinations is to address the lead-lag characteristics sin and cos. In addition, it is worth mentioning that the non-synchronous nature of the tan function will produce asymmetric values when it combines with the sin or cos. Thus, this article has experimented with the standalone usage of the tan in the exploration and exploitation phases. The range variations in the trigonometric functions are carried out using the MOP coefficient.

The term ATOAsc represents that the sin function is used in the exploration phase and the cos function during the exploitation phase. Thus, the position update of the ATOAsc algorithm during exploration and exploitation phases is given as follows: (15)$$x_{i,j}\left(t+1\right)=\left\{\begin{array}{cc} best\left(x_{j}\right)÷\left(\mathrm{MOP}+\epsilon \right)\;\times \;\left(\sin\left({UB}_{j}-{LB}_{j}\right)\;\times \;\mu +\sin\left({LB}_{j}\right)\right), & r_{2}\leq 0.5,\\ best\left(x_{j}\right)\;\times \;\mathrm{MOP}\;\times \;\left(\sin\left({UB}_{j}-{LB}_{j}\right)\;\times \;\mu +\sin\left({LB}_{j}\right)\right), & r_{2}>0.5, \end{array}\right.$$
(16)$$x_{i,j}\left(t+1\right)=\left\{\begin{array}{cc} best\left(x_{j}\right)-\mathrm{MOP}\;\times \;\left(\cos\left({UB}_{j}-{\mathrm{LB}}_{j}\right)\;\times \;\mu +\cos\left({LB}_{j}\right)\right), & r_{3}\leq 0.5,\\ best\left(x_{j}\right)+\mathrm{MOP}\;\times \;\left(\cos\left(UB_{j}-LB_{j}\right)\;\times \;\mu +\cos\left(LB_{j}\right)\right), & r_{3}>0.5. \end{array}\right.$$

Similarly, in ATOAcs, the cos function will be automatically assigned as the initial function during the exploration phase and the sin operation in the exploitation phase. Thus, the position update of this ATOAcs algorithm is given as follows: (17)$$x_{i,j}\left(t+1\right)=\left\{\begin{array}{cc} best\left(x_{j}\right)÷\left(\mathrm{MOP}+\epsilon \right)\;\times \;\left(\cos\left({UB}_{j}-{LB}_{j}\right)\;\times \;\mu +\cos\left({LB}_{j}\right)\right), & r_{2}\leq 0.5,\\ best\left(x_{j}\right)\;\times \;\mathrm{MOP}\;\times \;\left(\cos\left({UB}_{j}-{LB}_{j}\right)\;\times \;\mu +\cos\left({LB}_{j}\right)\right), & r_{2}>0.5, \end{array}\right.$$
(18)$$x_{i,j}\left(t+1\right)=\left\{\begin{array}{cc} best\left(x_{j}\right)-\mathrm{MOP}\;\times \;\left(\sin\left({UB}_{j}-{\mathrm{LB}}_{j}\right)\;\times \;\mu +\sin\left({LB}_{j}\right)\right), & r_{3}\leq 0.5,\\ best\left(x_{j}\right)+\mathrm{MOP}\;\times \;\left(\sin\left(UB_{j}-LB_{j}\right)\;\times \;\mu +\sin\left(LB_{j}\right)\right), & r_{3}>0.5. \end{array}\right.$$

In Equations (Equation 13) to (Equation 18),

$x_{i,j}\left(t+1\right)$ represents the $i^{th}$ solution in the ${\left(t+1\right)}^{th}$ iteration at the $j^{th}$ position.$best(x_{j})$ is best solution obtained at $j^{th}$ position.${UB}_{j}$ and ${LB}_{j}$ are the upper and lower boundaries at $j^{th}$ position.ϵ is the constant integer.μ is the search control parameter.

Table 1 gives the list of developed algorithms and the functions used during the exploration and exploitation phases. The implementation of all the developed algorithms is the same, and only the functions in the corresponding position update equation will change. For an example case, the pseudocode for the implementation of the proposed ATOAcs algorithm is described in Algorithm 1. In this case, the exploration phase uses the cos function, and the exploitation phase uses the sin function. On the other hand, during the standalone implementation of ATOAs, the sin function will be used in the exploration and exploitation phases.
**Algorithm 1** Pseudocode of ATOAcs1:Initialize *X*, *N*, *t*, *T*, min, max, α, μ, and ϵ.2:**while** t < T **do**3:   Calculate the $best(x_{j})$, MOA(t) and MOP(t).4:   **for** i=1:size(X,1) **do**5:     **for** j=1:size(X,2) **do**6:        Generate $r_{1}\in \left[0,1\right]$7:        **if** $r_{1}$≤ MOA **then**8:          Generate $r_{2}\in \left[0,1\right]$.9:          **if** $r_{2}$≤ 0.5 **then**10:             Compute the $x_{i,j}\left(t+1\right)$ using the first case in (Equation 17).11:          **else**12:             Compute the $x_{i,j}\left(t+1\right)$ using the second case in (Equation 17).13:          **end if**14:        **else**15:          Generate $r_{3}\in \left[0,1\right]$.16:          **if** $r_{3}$≤ 0.5 **then**17:             Compute the $x_{i,j}\left(t+1\right)$ using the first case in (Equation 18).18:          **else**19:             Compute the $x_{i,j}\left(t+1\right)$ using the second case in (Equation 18).20:          **end if**21:        **end if**22:     **end for**23:   **end for**24:   t=t+125:**end while**26:**return**  *best(x_j_)*

## 3. Performance Analysis on Benchmark Functions

This section discusses the simulation and comparative analysis of the proposed ATOA algorithm with conventional AOA. Simulations over 33 different benchmark functions are carried out to validate the proposed ATOA efficiency. The corresponding benchmark functions and their equations and the desired range limits are given in Table 2. Here, the abbreviated terms cat. and func. represents the category and benchmark function, respectively. In all the cases, the different variants of the proposed ATOA algorithm and the existing AOA algorithm are compared to prove the effectiveness of the proposed algorithm. Furthermore, all the simulations are performed in MATLAB/Simulink software (2021a) using the 3.10 GHz Intel(R) Xeon PC with 16.00 GB of RAM.

### 3.1. Selection of Benchmark Functions

For effective comparison analysis, the benchmark functions are selected based on the independence of one another in multiple dimensions, different local minima, and diverse boundary values. The number of iterations and the population size of the search solutions are also kept constant, respectively, at 300 and 30. During the simulation analysis, the optimization parameters used in all the compared algorithms are min = 0.2, max = 1.0, α = 5, μ = 0.499, and $\epsilon =2.2204\;\times \;10^{-16}$. A single global minima value function is known as the unimodal function. Therefore, they are employed to test the exploitation precision capabilities of the optimization algorithm. Such unimodal benchmark functions used for the simulation are F1–F6, as given in Table 2.

The proposed method is also validated using the multimodal and hybrid composition functions in single and multidimension. The multimodal functions have multiple local minima points used to test the algorithms’ exploration ability, i.e., to check the ability to switch from the local minima to the global minima without getting caught in the same position. F7–F10 are the multimodal benchmark functions, and their equations are given in Table 2. In addition, the functions F11–F18 shown in Table 2 have multiple local minima but with fixed dimensions. These functions will help to test the stability of the algorithm. It is worth mentioning that the first 13 benchmark functions use higher dimension values of (30/100/500/1000) to evaluate the proposed ATOA. Other considered benchmark functions F19–F33 are hybrid, as given in Table 2. In the unimodal functions, the probability of getting caught in the same search space is minimal, as the optimal search solutions have a much clearer position update to move from their current position to the desired position. Additionally, in the case of hybrid functions from F19–F33, the search space identification and the next position movement are more challenging due to multiple local minima values.

The surface plots of all the considered functions are shown in Figure 4. These plots will help to visualize required search space requirements for the single and multi global minima functions. In these functions, many categories were present based on their surface search area and shape. The classifications are bowl-shaped, plate-shaped, valley-shaped, single, and multiple-local minima functions. These functions have multiple local minima values with an ample search space, wide range, numerous layers, and multidimensional characters.

### 3.2. Numerical Analysis on Benchmark Functions

In this section, the different variants of the proposed ATOA algorithm are numerically compared with the conventional AOA technique. The comparison results of the various benchmark functions (F1–F33) are given in Table 3. Here, the abbreviated terms refer to the different combinations of the trigonometric functions as mentioned in the methodology section. They are ATOAs, ATOAc, ATOAt, ATOAsc, and ATOAcs, which refer to ATOAsine, ATOAcos, ATOAtan, ATOAsinecos, and ATOAcossine, respectively. The performance of all the algorithms is evaluated in terms of mean value, the best solution, worst value, and the standard deviation (std. dev.).

The obtained results have shown a tough competition between the proposed ATOAcs and the AOA in the unimodal benchmark functions (F1–F6) in the mean, best, and worst value. During this performance, the proposed ATOAcs obtained significantly closer convergence to the global minima. While noticing the standard deviation results of the unimodal function, both the ATOAcs and AOA had the balanced solution-attaining ability. Among the different proposed variants, the ATOAcs had better performance than AOA due to the leading characteristics of the cos function. The other proposed combinations had a performance setback while compared with the AOA in the unimodal functions. In the multimodal and the fixed-dimension benchmark functions (F7–F18), all the proposed ATOA variants outperformed the conventional AOA in all the parameters, and among these, ATOAcs took the first position. This performance shows the efficiency of the proposed ATOA in the exploration and exploitation capabilities, which resulted in converging very closely or strictly at global minima value. It is worth mentioning that in the performance in the hybrid benchmark functions, the proposed ATOA variants produced the same results as the previous stage with a higher and faster convergence rate.

Additionally, in order to compare the performance of the different techniques quantitatively, the Friedman ranking test is used [45]. This method ranks the functions based on the nonparametric analysis of variance produced in one complete run of the algorithm. The Friedman ranking test for the different compared algorithms for all the benchmark functions is given in Table 4. The ranking is based on the best mean value obtained by each algorithm in the respective function. For instance, rank 1 denotes the closest value to the global minima, and rank 6 denotes the farthest convergence value.

In the initial benchmark functions, the existing AOA had dominance, but in the latter, the proposed ATOA variants showed outstanding search space identification, position update, and convergence performance. Regarding the final mean values, the ATOAcs had the least mean value of 2.454 and secured the first rank. The conventional AOA had a mean value of 3.696 that was 1.242 times higher than ATOAcs. ATOAs took second place by obtaining the value of 2.636. The respective values of ATOAsc and ATOAc were 3.242 and 2.878. The proposed ATOAt had a value of 3.484, which is almost close to the conventional AOA. Thus, the proposed technique validates their abilities to attain the best mean value for the numerical analysis, and the same results are reflected in the convergence performance.

### 3.3. Convergence Analysis

The convergence speed plot of the different algorithms in the benchmark functions are shown in Figure 5. In the initial benchmark functions (F1–F6), the ATOA had a slow search rate due to the distribution of the obtained solutions to the different search regions instead of aggregating them in the current search area. However, the proposed ATOA overcomes those issues in the benchmark functions F7–F10 and shows faster convergence at the global minima with fewer iterations. In addition, in functions F11–F18, the proposed ATOA variants can converge all over the iterations continuously. This indicates that the proposed combinations of the trigonometric functions in this research enhance the exploration and exploitation search space of the original AOA.

The proposed trigonometric combinations assisted the existing AOA to tackle all issues by making the ATOA search mechanism faster, using every iteration’s current best optimal value. These made the proposed ATOA algorithms find those global minima values in lower iterations than the existing AOA algorithm in most benchmark functions. It is worth mentioning that, in the hybrid benchmark functions F19–F29, ATOA found the optimal positions quickly even though these functions contain multiple local minima. The benchmark functions results of F20, F21, and F30–F33 show that the proposed ATOA experienced multiple local minima positions. However, the ATOA recovered quickly from those local minima positions and found the desired global minima with few iterations using the proposed trigonometric functions. The proposed ATOA accomplished better accuracy in most of the benchmark functions as opposed to the compared algorithms. Finally, sometimes the AOA is more effective than the ATOA in some of the initial functions. The overall ranking and the performance of the proposed ATOA combinations produced a better optimal solution.

## 4. Performance Analysis on Control of Real-time Pressure Process Plant

In the first part of this section, detailed information about the real-time pressure process plant, including its schematics and P&I diagram, is given, along with its mathematical modeling design procedure. Then, the selection of the controller and its implementation along with their optimized controller parameters are given. Finally, the performance analysis on the process plant is discussed.

### 4.1. Industrial-Scale Setup of Real-Time Pressure Process Plant

Figure 6a shows the experimental setup of the pressure process plant with its peripherals. The tank VL 202 is the main process tank supplied with a maximum pressure of 10 bar from the centralized air compressor. The hand valve HV 202 acts as the primary input pressure control for the process plant. Once this hand valve is opened, the input pressure to the buffer tank is controlled using the process control valve PCV 202. The pressure inside the VL 202 is measured using the digital pressure transmitter PT 202 attached to the process tank. The PT 202 measures the pressure values and converts them to digital voltage signals in the range of 0 to 5 V. The pressure-indicating controller PIC 202 receives these voltage signals and feeds them to the host PC via I/O interface boards to generate the control signal for the PCV 202. Additionally, an analog pressure gauge is available to indicate the real-time pressure changes inside the tank.

The process tank contains a pressure-release hand valve attached at the bottom to release the pressure during an emergency. This hand valve provides a safer way to discharge the pressurized air from the VL 202 in case of a pneumatic process control valve (PCV 202) failure. The pressure inside the buffer tank is regulated by releasing the excess air via a gas outlet available on top of the process tank. This outlet gas pipe is connected with another process control valve, PCV 203. The PCV 303 is maintained at 50% opening during the experimentation to avoid excessive pressure build-up inside the VL 202. The PCV 202 and PCV 203 receive their control signals from the host PC via the I/O interface boards.

The piping and instrumentation diagram of the pressure process plant is illustrated in Figure 6b. The pressure process plant is controlled using “Remote Desktop Connection” mode from the central control room for safety. The host PC in the process plant uses the MATLAB/Simulink software to generate control signals based on the input set-point and the actual process variable. Data transmission and communication between the host PC and the process field devices (i.e., control valve actuator and pressure transmitter) happens via the peripheral component interconnect (PCI) cards. The PCI card contains three separate modules, namely 1713U, 1720U, and 1751. PCI-1720U performs the function of sending the control signals to the host PC using its 12-bit, 4-channel analog output port. In addition, all the interfacing cards are provided with isolation protection of 2500 V DC between the outputs of the PCI bus. The analog input from the process plant is received via a 12-bit, 32-channel analog input card PCI-1713U with a sampling rate of 100 k samples. The PCI-1751 card carries out the digital signals’ data transmission from the PT 202 to the host PC and vice versa. It contains 48 bits of parallel digital input/output, enabling the process plant to be controlled remotely.

The mathematical modeling of the process plant is obtained using the open-loop step response characteristics. This response contains all the behavioral data and process dynamics such as process gain *K*, process dead-time $L_{p}$, and process time constant *T*. Thus an open-loop step response of the pressure process is carried out, and the plant dynamics are obtained as follows: K=0.866, $L_{p}=1$, and T=1.365. Based on the first-order plus dead-time system characteristic equation, the process model of the pressure process plant is constructed using the obtained plant dynamics. Thus, the transfer function of the pressure process plant is given as follows:(19)$$G_{p}\left(s\right)=\frac{K}{1+Ts}e^{-sL_{p}}=\frac{0.866}{1+1.365s}e^{-s}.$$

### 4.2. ATOA-Based Fractional-Order Predictive PI Control of Pressure Process Plant

PI controllers are the most well-known and widely adopted controllers in most industries. However, they perform poorly in an environment having delay, noise, network-induced delay, and high-frequency noise [47,48]. On the other hand, advanced control strategies, such as model predictive controller and generalized predictive controller, are computationally and structurally complex for implementation. Thus, the controllers need a simple design, effective dead-time compensation, and high disturbance-rejection characteristics to overcome the above issues. Such a type of controller is proposed by Arun et al. [46], which has the dead-time compensating capability of the Smith predictor and robustness nature of the fractional-order controllers. Therefore, in this article, such an efficient fractional-order predictive PI (FOPPI) controller designed by them is utilized. The selection of pressure process application for this research is due to its nonlinearity and sensitivity [49]. The FOPPI controller generates an effective control signal which is free from load variations and plant uncertainties. However, their controller parameters are obtained analytically, and those values are insufficient to mitigate the real-time process disturbances [50]. Thus, the proposed ATOA and conventional AOA techniques are utilized in this research to obtain effective controller parameters. The adequately tuned parameters are used for experimentation on the real-time pressure process plant.

The control signal u(s) of the adopted FOPPI controller is given as
(20)$$u\left(s\right)=K_{p}\left(1+\frac{1}{T_{i}s^{\lambda }}\right)e\left(s\right)-\frac{1}{T_{i}s^{\lambda }}\left(1-e^{-sL_{p}}\right)u\left(s\right),$$
where $K_{p}=\frac{1}{K}$, and $T_{i}=T$. In addition, u(s) and e(s) are the control and error signals, $K_{p}$ is the proportional gain, $T_{i}$ is the integral time, λ is the fractional-order integrator, and $L_{p}$ is the process dead-time.

For the optimization algorithms, the objective is to obtain the effective controller parameters (i.e., $K_{p}$, $K_{i}$, and λ), the $K_{i}$ is the integral gain of the controller, and it is obtained by $K_{i}=\frac{K_{p}}{T_{i}}$. In the analytical design technique, the FOPPI controller parameters are obtained directly from its design. The closed-loop system containing the FOPPI controller and the obtained pressure process model $G_{p}\left(s\right)$ in the presence of the ATOA is shown in Figure 7. The proposed ATOA is used to tune the FOPPI controller parameters using the integral time absolute error (ITAE) value. The ITAE value is assigned as the objective function for the proposed ATOA and AOA. The calculation of the ITAE value is obtained as
(21)$$\mathrm{ITAE}=\int_{0}^{\infty }t\left|e\left(t\right)\right|dt.$$

### 4.3. Performance Analysis

The performance analysis of the FOPPI under various types of ATOA and AOA optimizations are compared in terms of process rise time ($t_{r}$), settling time before ($t_{s1}$) and ($t_{s2}$), overshoot (%OS), and disturbance rejection. A 30% disturbance is injected at 150 s in the process feedback loop to investigate the set-point tracking capability of the FOPPI controller. In the analytical design, the controller parameters are obtained based on the FOPPI design structure itself. The different optimized FOPPI controllers using the various optimization algorithms and their numerical performance analysis are given in Table 5.

While observing the numerical analysis results given in Table 5, the proposed ATOA algorithm variants performed better than the AOA and the analytical design. Among all the compared algorithms, the existing analytical method had the faster rise time of 0.7665, followed by AOA, ATOASC, ATOAt, ATOAs, ATOAc, and ATOAcs, respectively. In this, the first-ranked ATOAcs had the slowest rise time, of 2.4432 s, compared to other methods. The proposed ATOAcs settled faster than all at 61.3744 s before the disturbance injection, even with the slowest rise time. Second place is secured by ATOAs with the settling time at 65.8043 s, followed by ATOAsc at 69.5402 s, ATOAc at 72.1039 s, AOA at 75.8397 s, ATOAt at 77.8114s, and analytical design last with 83.0137s. In this performance, the Friedman ranking order of the algorithms is also reflected, except for ATOAt.

After the disturbance injection, the AOA and analytical design managed to track the set-point, but both of them settled slower than the proposed ATOA at 276.7360 s and 280.4273 s, respectively. Here, the ATOAt managed to settle faster than the AOA and analytical design with a difference of 2.7189 s and 6.4102 s, respectively. In this case, the ATOAcs recovered effectively from the disturbance and settled 22.7199 s faster than the analytical design. The controller parameters obtained by the proposed algorithms significantly reduced the peak overshoot values, which indicates the excellent objective function-finding capability of the ATOA techniques. In the peak overshoot performance, ATOAc produced the most negligible value of 2.8020%, which is 19.427% less than the analytical design. ATOAt had the second-least value of 3.1546%, followed by ATOAs, ATOAsc, ATOACS, AOA, and analytical technique.

The disturbance rejection, peak overshoot, and set-point tracking performance of the compared algorithms are shown in Figure 8. The obtained results indicate the excellent tracking and faster recovering ability of the proposed ATOA optimized controller parameters. Zoomed regions of Figure 8 during the before and after disturbance are given in Figure 9. Regarding the results in region A, the AOA and analytical method produced more oscillations, which will significantly deteriorate the lifetime and performance of the control valve actuator. The same trend is repeated during the performance after the disturbance injection (see region B). It is worth mentioning that the control signals generated from the ATOA optimized values are less aggressive, and smooth compared with AOA and analytical design (see regions C and D). Once again, a similar pattern of Friedman ranking results is obtained in the ITAE performance, and the ATOAcs had the minimal value of 1.4224. Then, ATOAs had the second-least value of 1.7391, followed by ATOAsc with 2.0154. Here, the conventional AOA almost had a comparable value to the proposed ATOAt. Lastly, the analytical design had a value of 3.7159, 2.2935 higher than the proposed ATOAcs.

## 5. Summary and Conclusions

In the initial part of this section, the contributions of the proposed research are summarized. Then, the concluding remarks and the future directions of the research are provided.

### 5.1. Summary

The different combinations of the proposed ATOA, AOA, and the conventional analytical design algorithms helped us to investigate their performance on various benchmark functions and the real-time pressure process. The usage of the basic arithmetic and trigonometric functions of the ATOA provides a straightforward design and easy implementation. However, the advanced functions such as modulus, cubic, and sigmoid may catch up with the proposed technique’s performance. At the same time, even after the addition of the new functions, the proposed ATOA maintains the simple structure of the conventional SCA and AOA. All the above claims are validated based on the obtained results, which shows the faster convergence ability of ATOA in lesser iterations. Furthermore, the Friedman ranking of the compared algorithms given in Table 4 validates the effectiveness of using the trigonometric functions in the existing algorithms. In addition, the ATOA technique dramatically improves the search mechanism, which leads to the identification of the efficient, optimized FOPPI controller parameters shown in Table 5. The following points highlight some of the essential contributions of this article:1.The proposed ATOA outperformed all the compared algorithms in most of the benchmark functions in terms of mean, best, and standard deviation.2.The proposed ATOA variants produced the best global minima in fewer number of iterations. Among them, ATOAcs achieved phenomenal results in all the comparative research analysis.3.The proposed ATOAcs and ATOAs had an efficient global optima search mechanism and yielded better performance, and it is proven by the Friedman ranking test given in Table 4.4.The ATOA-optimized FOPPI controller parameters performed effectively by reducing the peak overshoot, actively tracking the set-point, and efficiently minimizing the disturbance impacts on the process.5.The control signals of the optimized FOPPI controller greatly smooth the control actions by filtering out the undesired stochastic disturbances.

### 5.2. Conclusions

This paper develops an enhanced arithmetic–trigonometric optimization algorithm by incorporating different trigonometric functions, namely sin, cos, and tan. The performance of the proposed metaheuristic ATOA algorithm was validated on thirty-three different optimization benchmark functions. Furthermore, the ATOAs were compared for the mean, global best, worst, and standard deviation performance to validate their effectiveness. The convergence performance results showed the proposed ATOA to achieve faster global minima in fewer iterations. Friedman ranking of the comparison of the different algorithms showed the improvements obtained by the proposed ATOA algorithm. Finally, experimentation on the real-time pressure process was carried out to prove the dynamic abilities of the proposed algorithm in obtaining the best optimal solution. In future, newer evolutionary and additional arithmetic operators will be considered. Furthermore, an attempt to hybridize the proposed ATOA with the other population-based metaheuristic optimization algorithms will be made.

## Figures and Tables

**Figure 1 sensors-22-00617-f001:**
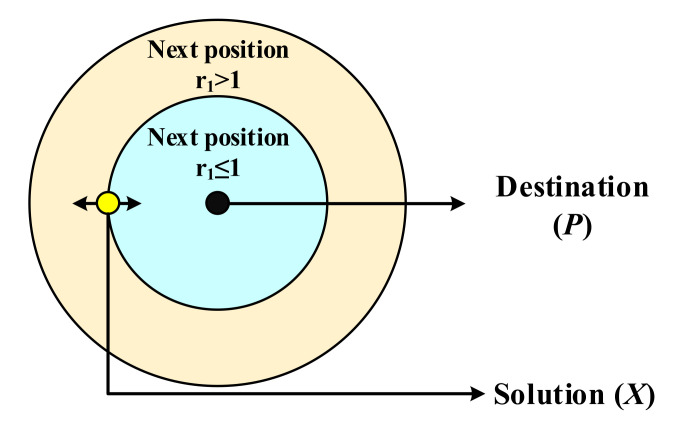
Position update mechanism in SCA algorithm.

**Figure 2 sensors-22-00617-f002:**
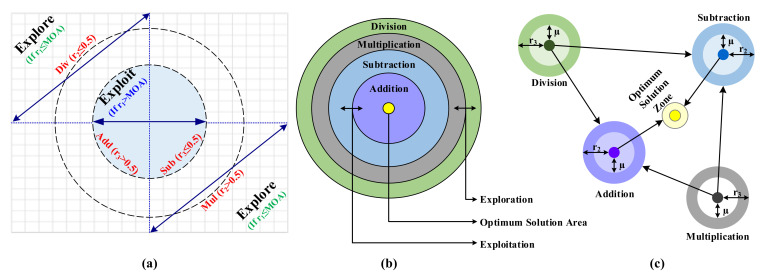
Illustration of AOA algorithm; (**a**) search phases of the AOA; (**b**) hierarchy of the arithmetic operators; (**c**) position update towards the optimum area.

**Figure 3 sensors-22-00617-f003:**
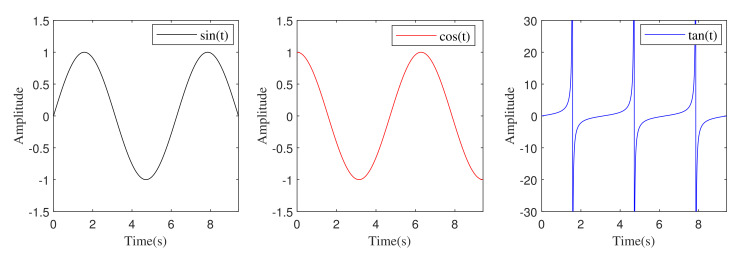
Different trigonometric functions plot in the range of [0,3π].

**Figure 4 sensors-22-00617-f004:**
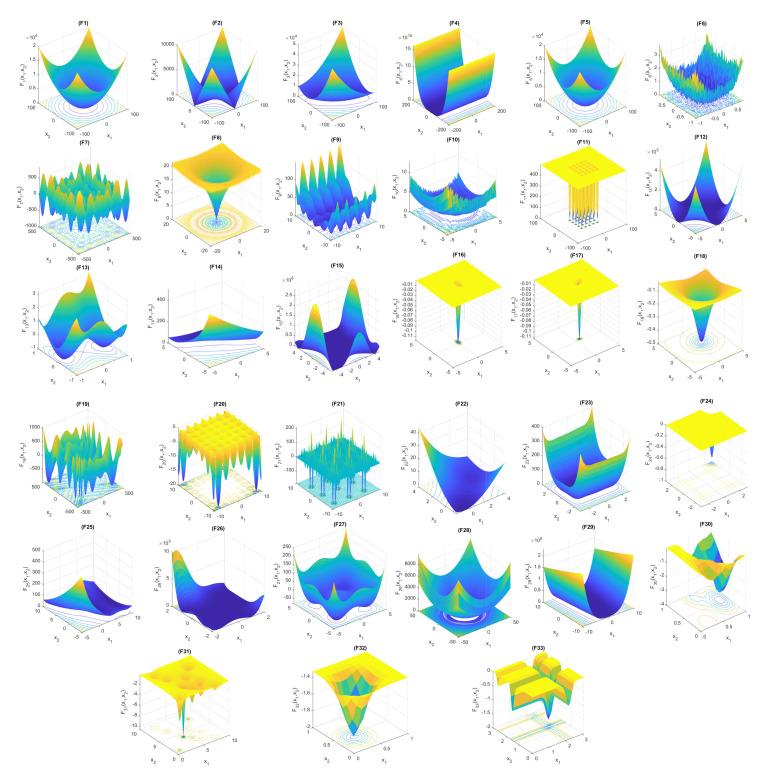
Search space plots of the benchmark functions.

**Figure 5 sensors-22-00617-f005:**
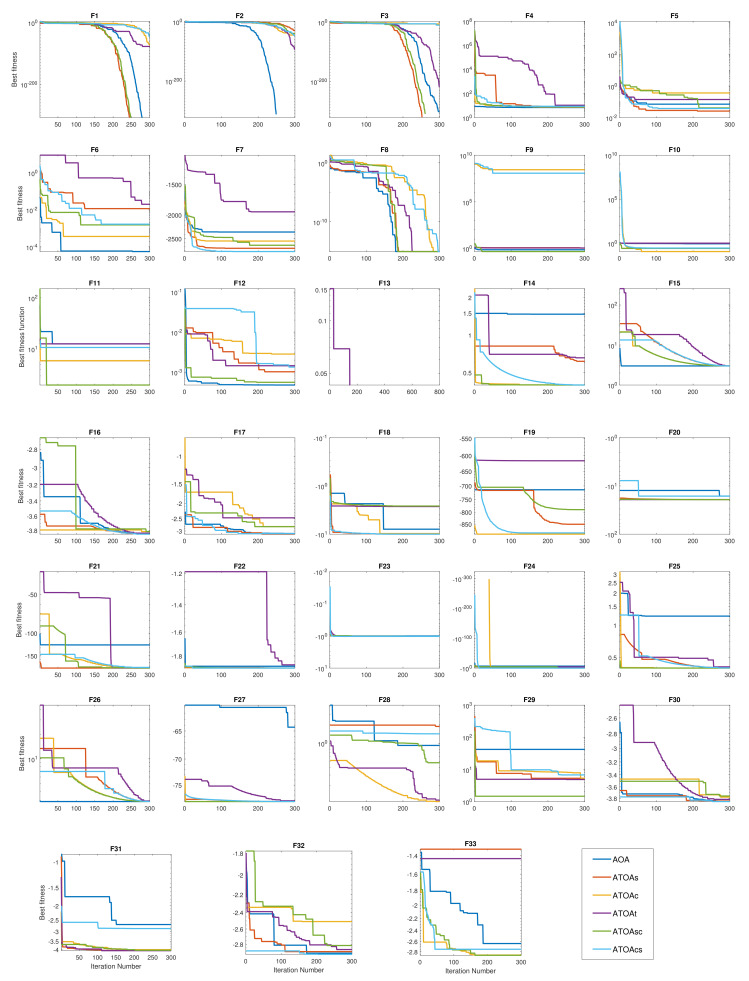
Convergence plots obtained from the different variants of the ATOA algorithms on various benchmark functions.

**Figure 6 sensors-22-00617-f006:**
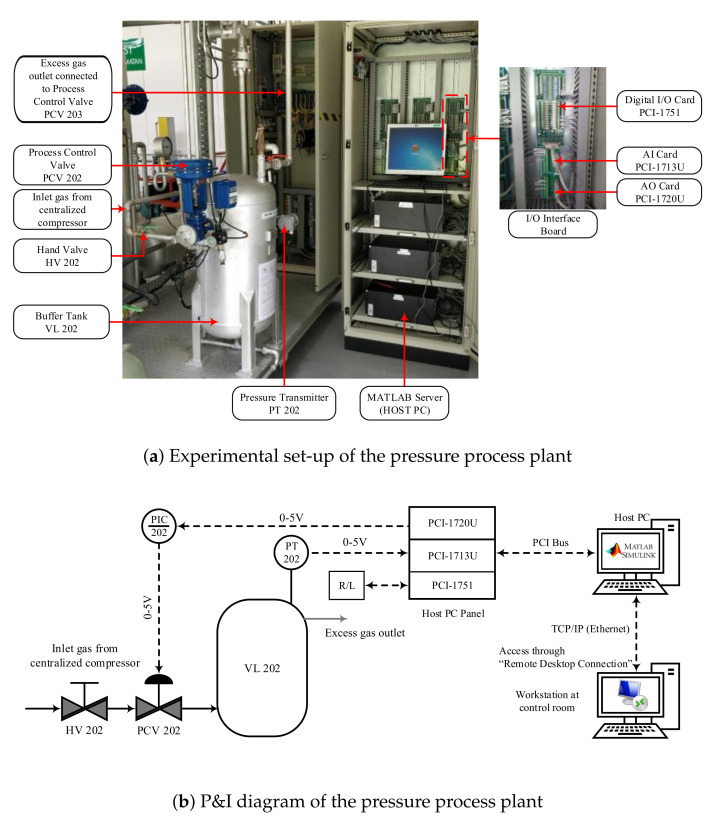
Schematic representation of the pressure process plant [46].

**Figure 7 sensors-22-00617-f007:**
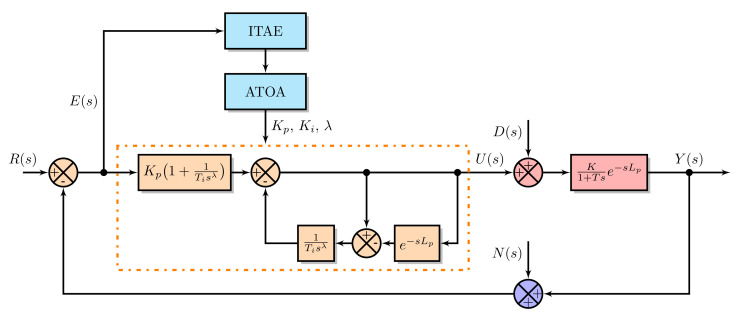
Closed-loop control system with the FOPPI controller in presence of ATOA.

**Figure 8 sensors-22-00617-f008:**
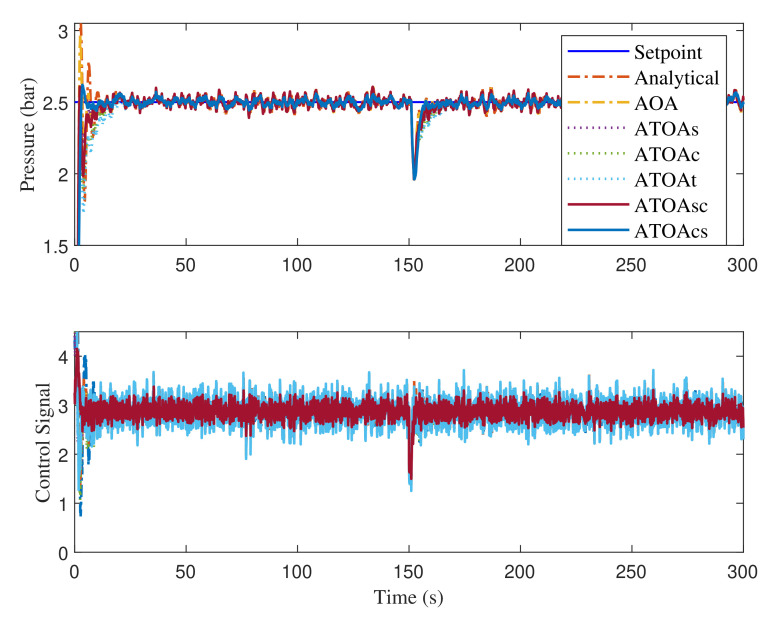
Disturbance rejection and set-point tracking performance of the FOPPI controller.

**Figure 9 sensors-22-00617-f009:**
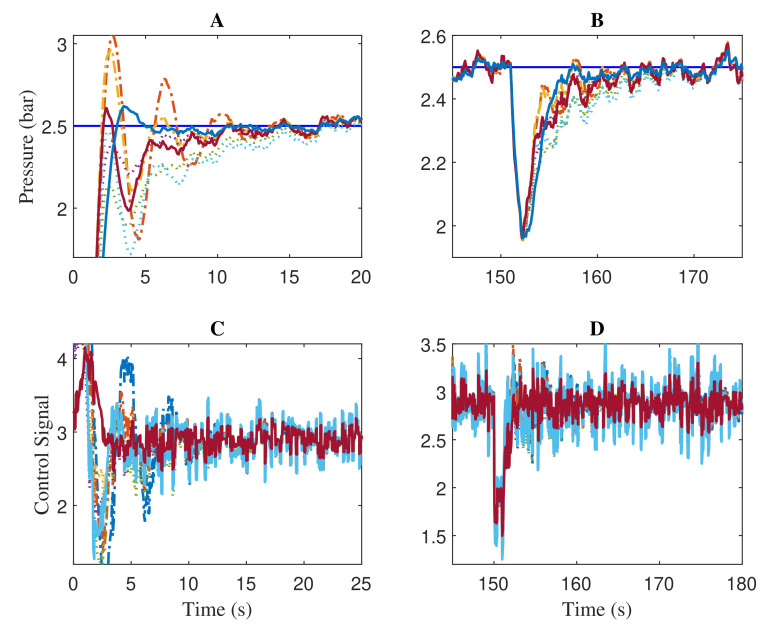
Zoomed regions of Figure 8 showing the process output (**A**,**B**) and control action (**C**,**D**).

**Table 1 sensors-22-00617-t001:** List of developed algorithms with functions used in exploration and exploitation phases.

Algorithm	Exploration Function	Exploitation Function
ATOAs	sin	sin
ATOAc	cos	cos
ATOAt	tan	tan
ATOAsc	sin	cos
ATOAcs	cos	sin

**Table 2 sensors-22-00617-t002:** Considered benchmark functions for the performance analysis.

Cat.	Func.	Description	Range
Unimodal	F1	$F\left(x\right)=\sum_{i=1}^{n}x_{i}^{2}$	[−100, 100]
	F2	$F\left(x\right)=\sum_{i=0}^{n}\left|x_{i}\right|+\left[r\right]\prod_{i=0}^{n}\left|x_{i}\right|$	[−10, 10]
	F3	$F\left(x\right)=\sum_{i=1}^{d}{\left(\sum_{j=1}^{i}x_{j}\right)}^{2}$	[−100, 100]
	F4	$F\left(x\right)=\sum_{i=1}^{n-1}\left[100{\left(x_{i}^{2}-x_{i+1}\right)}^{2}+{\left(1-x_{i}\right)}^{2}\right]$	[−30, 30]
	F5	$F\left(x\right)=\sum_{i=1}^{n}{\left(\left[x_{i}+0.5\right]\right)}^{2}$	[−100, 100]
	F6	$F\left(x\right)=\sum_{i=0}^{n}ix_{i}^{4}+\mathrm{random}\left(0,1\right)$	[−128, 128]
Multimodal	F7	$F\left(x\right)=\sum_{i=1}^{n}\left(-x_{i}\sin\left(\sqrt{\left|x_{i}\right|}\right)\right)$	[−500, 500]
	F8	$\begin{array}{c} F\left(x\right)=-20\exp\left(-0.2\sqrt{\frac{1}{n}\sum_{i=1}^{n}x_{i}^{2}}\right)-\exp\left(\frac{1}{n}\sum_{i=1}^{n}\cos\left(2\pi x_{i}\right)\right)+20+e \end{array}$	[−32, 32]
	F9	$\begin{array}{cc} F(x)= & \frac{\pi }{n}\left\{10\sin\left(\pi y_{1}\right)\right\}+\sum_{i=1}^{n-1}{\left(y_{i}-1\right)}^{2}[1+10\sin^{2}\left(\pi y_{i+1}\right)+\sum_{i=1}^{n}u\left(x_{i},10,100,4\right),\\ & \mathrm{where}\;y_{i}=1+\frac{x_{i}+1}{4},u\left(x_{i},a,k,m\right)\left\{\begin{array}{cc} K{\left(x_{i}-a\right)}^{m} & x_{i}>a\\ 0 & -a\leq x_{i}\geq a\\ K{\left(-x_{i}-a\right)}^{m} & -a\leq x_{i} \end{array}\right. \end{array}$	[−50, 50]
	F10	$\begin{array}{cc} F(x)= & 0.1\left(\sin^{2}\left(3\pi x_{1}\right)+\sum_{i=1}^{n}{\left(x_{i}-1\right)}^{2}\left[1+\sin^{2}\left(3\pi x_{i}+1\right)\right]+{\left(x_{n}-1\right)}^{2}\left[1+\sin^{2}\left(2\pi x_{n}\right)\right]\right)\\ & +\sum_{i=1}^{n}u\left(x_{i},5,100,4\right) \end{array}$	[−50, 50]
	F11	$F\left(x\right)={\left(\frac{1}{500}+\sum_{j=1}^{25}\frac{1}{j+\sum_{i=1}^{2}}\left(x_{i}-a_{ij}\right)\right)}^{-1}$	[−65,65]
	F12	$F\left(x\right)=\sum_{i=1}^{11}{\left[a_{i}-\frac{x_{1}\left(b_{i}^{2}+b_{i}x_{2}\right)}{b_{i}^{2}+b_{i}x_{3}+x_{4}}\right]}^{2}$	[−5, 5]
	F13	$F\left(x\right)=4x_{1}^{2}-2.1x_{1}^{4}+\frac{1}{3}x_{1}^{6}+x_{1}x_{2}-4x_{2}^{2}+4x_{2}^{4}$	[−5, 5]
	F14	$\begin{array}{c} F\left(x\right)={\left(x_{2}-\frac{5.1}{4\pi^{2}}x_{1}^{2}+\frac{5}{\pi }x_{1}-6\right)}^{2}+10\left(1-\frac{1}{8\pi }\right)\cos x_{1}+10 \end{array}$	[−5, 5]
	F15	$\begin{array}{c} F\left(x\right)=\sum_{i=1}^{d/4}\left[{\left(x_{4i-3}+10x_{4i-2}\right)}^{2}+5{\left(x_{4i-1}-x_{4i}\right)}^{2}+{\left(x_{4i-2}-2x_{4i-1}\right)}^{4}+10{\left(x_{4i-3}-x_{4i}\right)}^{4}\right] \end{array}$	[−4, 5]
	F16	$F\left(x\right)=-\sum_{i=1}^{4}c_{i}\exp\left(-\sum_{i=1}^{3}a_{ij}{\left(x_{j}-p_{ij}\right)}^{2}\right)$	[−1, 2]
	F17	$F\left(x\right)=-\sum_{i=1}^{4}c_{i}\exp\left(-\sum_{i=1}^{6}a_{ij}{\left(x_{j}-p_{ij}\right)}^{2}\right)$	[0, 1]
	F18	$F\left(x\right)=-\sum_{i=1}^{5}{\left[\left(X-a_{i}\right){\left(X-a_{i}\right)}^{T}+c_{i}\right]}^{-1}$	[0, 1]
Hybrid	F19	$\begin{array}{c} F\left(x\right)=-\left(x_{2}+47\right)\sin\left(\sqrt{\left|x_{2}+\frac{x_{1}}{2}+47\right|}\right)\\ -x_{1}\sin\left(\sqrt{\left|x_{1}-\left(x_{2}+47\right)\right|}\right) \end{array}$	[−512, 512]
	F20	$F\left(x\right)=-\left|\sin\left(x_{1}\right)\cos\left(x_{2}\right)\exp\left(\left|1-\frac{\sqrt{x_{1}^{2}+x_{2}^{2}}}{\pi }\right|\right)\right|$	[−10, 10]
Hybrid	F21	$\begin{array}{c} F\left(x\right)=\left(\sum_{i=1}^{5}i\cos\left(\left(i+1\right)x_{1}+i\right)\right)\;\times \;\left(\sum_{i=1}^{5}i\cos\left(\left(i+1\right)x_{2}+i\right)\right) \end{array}$	[−5.12, 5.12]
	F22	$F\left(x\right)=\sin\left(x_{1}+x_{2}\right)+{\left(x_{1}-x_{2}\right)}^{2}-1.5x_{1}+2.5x_{2}+1$	[−1.5, 4]
	F23	$F\left(x\right)=\left(4-2.1x_{1}^{2}+\frac{x_{1}^{4}}{3}\right)x_{1}^{2}+x_{1}x_{2}+\left(-4+4x_{2}^{2}\right)x_{2}^{2}$	[−3, 3]
	F24	$F\left(x\right)=-\cos\left(x_{1}\right)\cos\left(x_{2}\right)\exp\left(-{\left(x_{1}-\pi \right)}^{2}-{\left(x_{2}-\pi \right)}^{2}\right)$	[−100, 100]
	F25	$F\left(x\right)=a{\left(x_{2}-bx_{1}^{2}+cx_{1}-r\right)}^{2}+s\left(1-t\right)\cos\left(x_{1}\right)+s$	[−5, 10]
	F26	$\begin{array}{cc} F(x)= & \left[1+{\left(x_{1}+x_{2}+1\right)}^{2}\left(19-14x_{1}+3x_{1}^{2}-14x_{2}+6x_{1}x_{2}+3x_{2}^{2}\right)\right]\\ & \;\times \;\left[30+{\left(2x_{1}-3x_{2}\right)}^{2}\left(18-32x_{1}+12x_{1}^{2}+48x_{2}-36x_{1}x_{2}+27x_{2}^{2}\right)\right] \end{array}$	[−2, 2]
	F27	$F\left(x\right)=\frac{1}{2}\sum_{i=1}^{d}\left(x_{i}^{4}-16x_{i}^{2}+5x_{i}\right)$	[−5, 5]
	F28	$\begin{array}{cc} F(x)= & \sin^{2}\left(\pi w_{1}\right)+\sum_{i=1}^{d-1}{\left(w_{i}-1\right)}^{2}\left[1+10\sin^{2}\left(\pi w_{i}+1\right)\right]+{\left(w_{d}-1\right)}^{2}\left[1+\sin^{2}\left(2\pi w_{d}\right)\right],\\ & \mathrm{where}\;w_{i}=1+\frac{x_{i}-1}{4},\;\mathrm{for}\;\mathrm{all}\;i=1,\ldots ,d \end{array}$	[−10, 10]
	F29	$\begin{array}{cc} F(x)= & 100{\left(x_{1}^{2}-x_{2}\right)}^{2}+{\left(x_{1}-1\right)}^{2}+{\left(x_{3}-1\right)}^{2}+90{\left(x_{3}^{2}-x_{4}\right)}^{2}\\ & +10.1\left({\left(x_{2}-1\right)}^{2}+{\left(x_{4}-1\right)}^{2}\right)+19.8\left(x_{2}-1\right)\left(x_{4}-1\right) \end{array}$	[−10, 10]
	F30	$\begin{array}{cc} F(x)= & -\sum_{i=1}^{4}\alpha_{i}\exp\left(-\sum_{j=1}^{3}A_{ij}{\left(x_{j}-P_{ij}\right)}^{2}\right),\;\mathrm{where}\;\alpha ={\left(1.0,1.2,3.0,3.2\right)}^{T},\\ & A=\left(\begin{array}{ccc} 3.0 & 10 & 30\\ 0.1 & 10 & 35\\ 3.0 & 10 & 30\\ 0.1 & 10 & 35 \end{array}\right),\;P=10^{-4}\left(\begin{array}{ccc} 3689 & 1170 & 2673\\ 4699 & 4387 & 7470\\ 1091 & 8732 & 5547\\ 381 & 5743 & 8828 \end{array}\right). \end{array}$	[0, 1]
	F31	$\begin{array}{cc} F(x)= & -\sum_{i=1}^{m}{\left(\sum_{j=1}^{4}{\left(x_{j}-C_{ji}\right)}^{2}+\beta_{i}\right)}^{-1},\;\mathrm{where}\;m=10,\;\beta =\frac{1}{10}{\left(1,2,2,4,4,6,3,7,5,5\right)}^{T},\\ & \;C=\left(\begin{array}{cccccccccc} 4.0 & 1.0 & 8.0 & 6.0 & 3.0 & 2.0 & 5.0 & 8.0 & 6.0 & 7.0\\ 4.0 & 1.0 & 8.0 & 6.0 & 7.0 & 9.0 & 3.0 & 1.0 & 2.0 & 3.6\\ 4.0 & 1.0 & 8.0 & 6.0 & 3.0 & 2.0 & 5.0 & 8.0 & 6.0 & 7.0\\ 4.0 & 1.0 & 8.0 & 6.0 & 7.0 & 9.0 & 3.0 & 1.0 & 2.0 & 3.6 \end{array}\right). \end{array}$	[0, 10]
	F32	$\begin{array}{cc} F(x)= & -\sum_{i=1}^{4}\alpha_{i}\exp\left(-\sum_{j=1}^{6}A_{ij}{\left(x_{j}-P_{ij}\right)}^{2}\right),\;\mathrm{where}\;\alpha ={\left(1.0,1.2,3.0,3.2\right)}^{T},\\ & A=\left(\begin{array}{cccccc} 10 & 3 & 17 & 3.50 & 1.7 & 8\\ 0.05 & 10 & 17 & 0.1 & 8 & 14\\ 3 & 3.5 & 1.7 & 10 & 17 & 8\\ 17 & 8 & 0.05 & 10 & 0.1 & 14 \end{array}\right),\\ \;\;\;\;\;\;\;\;\;\;\;\;\;\;\;\;\;\;\;\; & P=10^{-4}\left(\begin{array}{cccccc} 1312 & 1696 & 5569 & 124 & 8283 & 5886\\ 2329 & 4135 & 8307 & 3736 & 1004 & 9991\\ 2348 & 1451 & 3522 & 2883 & 3047 & 6650\\ 4047 & 8828 & 8732 & 5743 & 1091 & 381 \end{array}\right). \end{array}$	[0, 1]
	F33	$F\left(x\right)=-\sum_{i=1}^{d}\sin\left(x_{i}\right)\sin^{2m}\left(\frac{ix_{i}^{2}}{\pi }\right)$	[0, π]

**Table 3 sensors-22-00617-t003:** Comparisons of the statistical results obtained from the different variants of the ATOA algorithms on various benchmark functions.

Function	Global Minima	Measure	AOA	ATOAs	ATOAc	ATOAt	ATOAsc	ATOAcs
F1	0	Mean	0.0000	4.46$\;\times \;10^{-33}$	9.89$\;\times \;10^{-28}$	0.0000	0.0000	0.0000
		Best	0.0000	0.0000	0.0000	0.0000	0.0000	0.0000
		Worst	0.0000	2.10$\;\times \;10^{-31}$	4.95$\;\times \;10^{-26}$	0.0000	0.0000	0.0000
		Std. Dev	0.0000	2.97$\;\times \;10^{-32}$	7.00$\;\times \;10^{-27}$	0.0000	0.0000	0.0000
F2	0	Mean	0.0000	8.65$\;\times \;10^{-139}$	4.13$\;\times \;10^{-86}$	1.67$\;\times \;10^{-241}$	5.95$\;\times \;10^{-55}$	0.0000
		Best	0.0000	0.0000	0.0000	0.0000	0.0000	0.0000
		Worst	0.0000	4.32$\;\times \;10^{-137}$	2.06$\;\times \;10^{-84}$	8.36$\;\times \;10^{-240}$	2.97$\;\times \;10^{-53}$	0.0000
		Std. Dev	0.0000	6.11$\;\times \;10^{-138}$	2.92$\;\times \;10^{-85}$	0.0000	4.21$\;\times \;10^{-54}$	0.0000
F3	0	Mean	0.0000	2.03$\;\times \;10^{-19}$	1.07$\;\times \;10^{-10}$	0.0000	0.0000	0.0000
		Best	0.0000	0.0000	0.0000	0.0000	0.0000	0.0000
		Worst	0.0000	1.02$\;\times \;10^{-17}$	5.35$\;\times \;10^{-9}$	0.0000	0.0000	0.0000
		Std. Dev	0.0000	1.44$\;\times \;10^{-18}$	7.57$\;\times \;10^{-10}$	0.0000	0.0000	0.0000
F4	0	Mean	5.1652	7.7668	6.9754	23.2905	6.8636	7.8511
		Best	4.6129	6.9030	5.2892	8.2527	5.2644	6.6216
		Worst	5.8078	8.1612	7.9103	547.483	8.4399	8.6583
		Std. Dev	0.2658	0.3346	0.499	79.9482	0.7056	0.4639
F5	0	Mean	0.0143	0.0018	0.3024	0.032	0.0726	0.0014
		Best	0.0054	0.0004	0.1694	0.0161	0.0279	0.0007
		Worst	0.0232	0.003	0.4172	0.0531	0.1604	0.0027
		Std. Dev	0.0041	5.23$\;\times \;10^{-4}$	0.056	0.0105	0.0318	0.0005
F6	0	Mean	0.0000	3.69$\;\times \;10^{-4}$	7.94$\;\times \;10^{-5}$	0.0211	6.11$\;\times \;10^{-4}$	0.0022
		Best	0.0000	2.91$\;\times \;10^{-5}$	3.22$\;\times \;10^{-6}$	7.45$\;\times \;10^{-4}$	5.37$\;\times \;10^{-5}$	0.0000
		Worst	0.0001	0.0018	2.29$\;\times \;10^{-4}$	0.0926	0.0018	0.0173
		Std. Dev	0.0000	3.83$\;\times \;10^{-4}$	5.79$\;\times \;10^{-5}$	0.0204	4.12$\;\times \;10^{-4}$	0.0029
F7	−418.9829 × n	Mean	−3475.0000	−3.22$\;\times \;10^{3}$	−3069.4000	−2.18$\;\times \;10^{3}$	−3.08$\;\times \;10^{3}$	−3178.2000
		Best	−4071.3000	−4.00$\;\times \;10^{3}$	−3670.2000	−3.04$\;\times \;10^{3}$	−3.58$\;\times \;10^{3}$	−3794.9000
		Worst	−2914.5000	−2.44$\;\times \;10^{3}$	−2341.0000	−1.36$\;\times \;10^{3}$	−2.52$\;\times \;10^{3}$	−2620.2000
		Std. Dev	252.5026	262.4675	284.0105	368.5892	228.423	311.1536
F8	0	Mean	0.0000	6.32$\;\times \;10^{-12}$	1.98$\;\times \;10^{-9}$	8.88$\;\times \;10^{-16}$	8.88$\;\times \;10^{-16}$	0.0000
		Best	0.0000	8.88$\;\times \;10^{-16}$	8.88$\;\times \;10^{-16}$	8.88$\;\times \;10^{-16}$	8.88$\;\times \;10^{-16}$	0.0000
		Worst	0.0000	3.16$\;\times \;10^{-10}$	9.91$\;\times \;10^{-8}$	8.88$\;\times \;10^{-16}$	8.88$\;\times \;10^{-16}$	0.0000
		Std. Dev	0.0000	4.47$\;\times \;10^{-11}$	1.40$\;\times \;10^{-8}$	0.0000	0.0000	0.0000
F9	0	Mean	0.5947	8.85$\;\times \;10^{6}$	5.88$\;\times \;10^{6}$	0.9595	0.4228	1.0271
		Best	0.4953	1.0361	3.2475	0.8689	0.3078	0.9451
		Worst	0.6443	1.67$\;\times \;10^{8}$	9.76$\;\times \;10^{7}$	1.0228	0.523	1.0944
		Std. Dev	0.0337	2.94$\;\times \;10^{7}$	1.52$\;\times \;10^{7}$	0.0315	0.0453	0.0317
F10	0	Mean	0.7823	0.1646	0.1116	0.9423	0.1327	0.8997
		Best	0.3881	0.0294	0.0554	0.5441	0.059	0.6860
		Worst	0.9948	0.4447	0.1734	0.9907	0.2184	0.9868
		Std. Dev	0.1563	0.1028	0.0245	0.0795	0.0376	0.0639
F11	1	Mean	8.8855	5.9519	5.5608	8.666	6.1167	5.6534
		Best	0.9980	0.998	0.998	0.998	0.998	0.9980
		Worst	12.6705	11.7187	12.6705	12.6705	12.6705	11.7187
		Std. Dev	3.9075	3.3638	3.1317	3.7149	3.6442	3.2853
F12	0.0003	Mean	0.0104	0.0016	0.0043	0.008	0.0111	0.0078
		Best	0.0003	4.02$\;\times \;10^{-4}$	3.92$\;\times \;10^{-4}$	6.66$\;\times \;10^{-4}$	3.13$\;\times \;10^{-4}$	0.0004
		Worst	0.0863	0.0245	0.0234	0.0569	0.0566	0.0572
		Std. Dev	0.0153	0.0037	0.0062	0.011	0.0133	0.0111
F13	−1.0316	Mean	−1.0316	−1.0313	−1.0316	−1.0238	−1.0316	−1.0313
		Best	−1.0316	−1.0316	−1.0316	−1.0311	−1.0316	−1.0316
		Worst	−1.0316	−1.0303	−1.0316	−1.0063	−1.0316	−1.0301
		Std. Dev	0.0000	3.27$\;\times \;10^{-4}$	9.91$\;\times \;10^{-6}$	0.0066	8.16$\;\times \;10^{-6}$	0.0003
F14	0.398	Mean	1.1631	0.4233	0.3979	0.4025	0.3979	0.4088
		Best	0.4123	0.3979	0.3979	0.398	0.3979	0.3979
		Worst	2.8698	0.7824	0.398	0.4356	0.3981	0.5690
		Std. Dev	0.5939	0.0758	2.16$\;\times \;10^{-5}$	0.0067	3.86$\;\times \;10^{-5}$	0.0259
F15	3	Mean	6.2400	3.0005	3.0002	8.3111	6.253	3.0005
		Best	3.0000	3.0000	3.0000	3.0000	3.0000	3.0000
		Worst	30.0000	3.0028	3.0006	91.5641	84.6034	3.0027
		Std. Dev	8.8630	6.54$\;\times \;10^{-4}$	1.50$\;\times \;10^{-4}$	21.2206	16.0971	0.0006
F16	−3.86	Mean	−3.8544	−3.8563	−3.8261	−3.8502	−3.8445	−3.8559
		Best	−3.8608	−3.8625	−3.854	−3.8616	−3.8548	−3.8614
		Worst	−3.8498	−3.8515	−3.0119	−3.8324	−3.8202	−3.8503
		Std. Dev	0.0023	0.0026	0.1181	0.0052	0.009	0.0024
F17	−0.32	Mean	−3.1259	−3.1647	−2.9368	−3.0409	−2.8466	−3.1710
		Best	−3.2643	−3.2551	−3.2132	−3.1812	−3.117	−3.2631
		Worst	−2.9339	−2.9039	−2.2038	−2.4088	−2.3154	−3.0584
		Std. Dev	0.0650	0.0595	0.1899	0.1378	0.2263	0.0473
F18	−10.1532	Mean	−4.1529	−8.2176	−7.7996	−6.5441	−5.3207	−4.7763
		Best	−8.0049	−10.1471	−10.0933	−10.1441	−10.1435	−10.1413
		Worst	−1.9980	−2.6275	−2.6046	−2.6249	−2.6102	−2.6253
		Std. Dev	1.2108	3.251	3.1242	3.2365	2.9748	2.7482
F19	−959.6407	Mean	−800.3974	−863.6167	−850.7201	−706.1923	−859.4495	−856.7580
		Best	−941.2958	−959.6407	−959.6407	−932.4704	−959.6406	−959.6407
		Worst	−644.2784	−559.7869	−559.7868	−474.3529	−545.6967	−575.2190
		Std. Dev	89.8151	97.5246	114.0678	116.7106	112.0762	86.4870
F20	−19.2085	Mean	−18.7179	−18.8544	−18.9728	−18.7377	−18.9140	−18.9721
		Best	−19.2083	−19.2084	−19.2085	−19.2085	−19.2085	−19.2084
		Worst	−15.8160	−16.2678	−16.2678	−16.2678	−16.2678	−16.2678
		Std. Dev	0.7836	0.9649	0.8057	1.0889	0.8910	0.8055
F21	−186.7309	Mean	−116.2292	−176.7813	−178.8541	−186.6622	−176.8742	−181.4309
		Best	−184.3297	−186.7222	−186.7259	−186.7258	−186.7211	−186.7294
		Worst	−50.6600	−79.3989	−123.4528	−186.5087	−64.6800	−123.0804
		Std. Dev	38.0811	25.0093	20.6675	0.0496	29.1740	17.3085
F22	−1.9133	Mean	−1.8773	−1.9132	−1.9132	−1.8676	−1.9132	−1.9131
		Best	−1.9132	−1.9132	−1.9132	−1.9128	−1.9132	−1.9132
		Worst	−1.4783	−1.9131	−1.9132	−1.6836	−1.9131	−1.9127
		Std. Dev	0.1158	3.84$\;\times \;10^{-6}$	3.64$\;\times \;10^{-7}$	0.0507	7.44$\;\times \;10^{-6}$	9.84$\;\times \;10^{-5}$
F23	−1.0316	Mean	−1.0091	−1.0316	−1.0314	−1.0316	−1.0314	−1.0316
		Best	−1.0316	−1.0316	−1.0316	−1.0316	−1.0316	−1.0316
		Worst	−0.9990	−1.0314	−1.0310	−1.0316	−1.0309	−1.0314
		Std. Dev	0.0135	4.68$\;\times \;10^{-5}$	1.50$\;\times \;10^{-4}$	3.78$\;\times \;10^{-9}$	1.65$\;\times \;10^{-4}$	4.25$\;\times \;10^{-5}$
F24	−1	Mean	−0.0532	−0.9400	−0.8598	−0.1601	−0.9998	−0.9638
		Best	−0.9963	−1.0000	−1.0000	−1.0000	−1.0000	−1.0000
		Worst	−2.92$\;\times \;10^{-9}$	−8.11$\;\times \;10^{-5}$	−8.09$\;\times \;10^{-5}$	−8.11$\;\times \;10^{-5}$	−0.9994	−0.9265
		Std. Dev	0.1934	0.2399	0.3504	0.3703	1.46$\;\times \;10^{-4}$	0.0169
F25	0.3978	Mean	0.7698	0.3989	0.3980	0.4073	0.3981	0.3993
		Best	0.3985	0.3979	0.3979	0.3982	0.3979	0.3979
		Worst	1.8135	0.4045	0.3984	0.4645	0.4000	0.4142
		Std. Dev	0.3683	0.0011	9.50$\;\times \;10^{-5}$	0.0103	4.03$\;\times \;10^{-4}$	0.0027
F26	3	Mean	21.7513	3.0005	3.0002	6.2811	3.0002	3.0004
		Best	3.0000	3.0000	3.0000	3.0000	3.0000	3.0000
		Worst	156.2760	3.0035	3.0010	85.6932	3.0008	3.0016
		Std. Dev	27.7673	7.45$\;\times \;10^{-4}$	2.02$\;\times \;10^{-4}$	16.2291	1.71$\;\times \;10^{-4}$	4.55$\;\times \;10^{-4}$
F27	−39.1659	Mean	−72.2967	−78.3312	−78.3323	−76.3317	−76.3532	−76.9175
		Best	−78.3276	−78.3323	−78.3323	−78.3312	−78.3323	−78.3323
		Worst	−61.4160	−78.3283	−78.3322	−64.1633	−64.1956	−64.1936
		Std. Dev	5.2366	8.32$\;\times \;10^{-4}$	2.35$\;\times \;10^{-5}$	4.9494	4.9551	4.2841
F28	0	Mean	1.4327	0.1547	0.1249	0.0135	0.0647	0.0955
		Best	0.2886	0.0016	6.65$\;\times \;10^{-5}$	2.44$\;\times \;10^{-5}$	3.00$\;\times \;10^{-5}$	3.75$\;\times \;10^{-4}$
		Worst	2.0000	0.6345	0.4463	0.1169	0.1157	0.1469
		Std. Dev	0.5954	0.1213	0.1332	0.0335	0.0548	0.0438
F29	0	Mean	38.8803	8.2303	5.4673	3.7098	3.2438	3.5159
		Best	6.4751	0.4230	0.1357	0.0792	0.2594	0.1630
		Worst	42.0000	35.4304	28.2788	8.2784	8.2392	8.8946
		Std. Dev	9.3207	8.8018	6.1918	3.0627	2.6992	2.6469
F30	−3.8627	Mean	−3.8545	−3.8560	−3.8128	−3.8366	−3.8257	−3.8558
		Best	−3.8603	−3.8621	−3.8576	−3.8616	−3.8562	−3.8612
		Worst	−3.8482	−3.8491	−3.0234	−3.0861	−3.0025	−3.8497
		Std. Dev	0.0029	0.0029	0.1620	0.1084	0.1194	0.0027
F31	−10.5364	Mean	−4.6271	−5.8656	−4.6245	−5.9376	−4.7527	−6.3397
		Best	−8.3487	−10.5298	−10.4165	−10.5222	−10.4922	−10.5194
		Worst	−2.1589	−1.8533	−1.8495	−1.8526	−1.6908	−1.8526
		Std. Dev	1.3407	3.6248	2.9955	3.2566	2.8037	2.4718
F32	−3.3223	Mean	−2.9401	−2.9589	−2.8212	−2.9047	−2.7873	−2.9603
		Best	−3.0060	−3.0201	−2.9447	−2.9750	−2.9337	−3.0257
		Worst	−2.8450	−2.8800	−2.5712	−2.7635	−2.5398	−2.8884
		Std. Dev	0.0292	0.0268	0.1079	0.0431	0.1204	0.0249
F33	−9.6601	Mean	−3.3874	−3.7494	−2.9064	−3.3626	−2.9680	−3.5913
		Best	−4.1525	−4.5626	−3.3311	−3.8498	−3.3714	−4.2671
		Worst	−2.6242	−2.6367	−2.3804	−2.3524	−2.5597	−2.7726
		Std. Dev	0.3532	0.4382	0.2460	0.3167	0.2131	0.3409

**Table 4 sensors-22-00617-t004:** Friedman ranking test for the optimization algorithms using the various benchmark functions.

Function	F1	F2	F3	F4	F5	F6	F7	F8	F9	F10	F11	F12
AOA	1	1	1	1	3	1	1	1	2	4	1	6
AOAs	2	3	2	4	2	3	2	2	6	3	4	1
ATOAc	3	4	3	3	6	2	5	3	5	1	6	2
ATOAt	1	2	1	6	4	6	6	1	3	6	2	4
ATOAsc	1	5	1	2	5	4	4	1	1	2	3	5
ATOAcs	1	1	1	5	1	5	3	1	4	5	5	3
**Function**	**F13**	**F14**	**F15**	**F16**	**F17**	**F18**	**F19**	**F20**	**F21**	**F22**	**F23**	**F24**
AOA	1	5	4	5	3	6	5	6	6	3	3	6
AOAs	2	4	2	1	2	1	1	4	5	1	1	3
AOAc	1	1	1	6	5	2	4	1	3	1	2	4
ATOAt	3	2	5	3	4	3	6	5	1	4	1	5
ATOAsc	1	1	3	4	6	4	2	3	4	1	2	1
ATOAcs	2	3	2	2	1	5	3	2	2	2	1	2
**Function**	**F25**	**F26**	**F27**	**F28**	**F29**	**F30**	**F31**	**F32**	**F33**	**Final Mean**	**Final Rank**	
AOA	6	5	6	6	6	3	5	3	6	3.696	6	
ATOAs	3	3	2	5	5	1	3	2	2	2.636	2	
ATOAc	1	1	1	4	4	6	6	5	5	3.242	4	
ATOAt	5	4	5	1	3	4	2	4	3	3.484	5	
ATOAsc	2	1	4	2	1	5	4	6	4	2.878	3	
ATOAcs	4	2	3	3	2	2	1	1	1	2.454	1	

**Table 5 sensors-22-00617-t005:** Performance comparison of the different algorithms in presence of FOPPI controller.

Algorithm	$K_{p}$	$K_{i}$	λ	$t_{r}$	$t_{s1}$	$t_{s2}$	%OS	ITAE
Analytical Design	1.150	0.842	0.98	0.7665	83.0137	280.4273	22.2290	3.7159
AOA	2.167	1.197	0.97	0.8543	75.8397	276.7360	18.0314	2.8243
ATOAs	1.987	1.056	0.99	1.0638	65.8043	261.9294	3.3561	1.7391
ATOAc	1.793	0.692	0.99	2.1729	72.1039	268.9153	2.8020	2.2007
ATOAt	1.983	0.592	0.98	0.9579	77.8114	274.0171	3.1546	2.6376
ATOAsc	2.321	1.025	0.99	0.8301	69.5402	264.9950	3.6644	2.0154
ATOAcs	1.321	0.897	0.98	2.4432	61.3744	257.7074	5.4593	1.4224

## Data Availability

Not applicable.

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
