# Peer review of "An Arithmetic-Trigonometric Optimization Algorithm with Application for Control of Real-Time Pressure Process Plant"

_sensors, 2022, doi:10.3390/s22020617_

Round 1
Reviewer 1 Report
Herein, the authors propose a new hybrid arithmetic-trigonometric optimization algorithm using different trigonometric functions for complex and continuously evolving real-time problems. The proposed algorithm adopts the trigonometric functions, namely sin, cos, and tan, with the conventional sine cosine algorithm and arithmetic optimization algorithm to improve the convergence rate and optimal search area in the exploration and exploitation phase. The article is interesting, merits publication in Sensors after taking into consideration the following minor corrections:
- Mention the cons/pros of the method.
- Mention the specification of the machine used to run the codes also, the platform used to write these codes.
- Comment on the efficiency, the complexity of the proposed algorithms.
- Proofread the whole manuscript for possible typos and grammatical errors.
Author Response
The authors appreciate the respective reviewer's positive feedback. All the feedbacks of the reviewer are seriously taken care of while preparing the revised manuscript. The response to the reviewer comment is in the attached PDF. also the highlighted response is available in the updated manuscript.

Reviewer 2 Report
This paper proposes a novel hybrid arithmetic-trigonometric optimization algorithm (ATOA) using different trigonometric functions for complex and continuously evolving real-time problems. This paper is well written and organized. Comments to the authors:
1) Highlight the contributions at the end of the Introduction section.
2) Explain the reason for using SCA algorithm over other algorithms reported in the literature.
3) How the math optimizer probability coefficient is calculated in this work?
4) Explain how the equations (13) and (14) are implemented in this work?
5) How the final rank in the results section has determined. Explain it in detail?
6) What is the impact of integral time absolute error (ITAE) value in the results section?
Author Response

(The authors gave the same response as above.)

Reviewer 3 Report
This paper proposes, "An Arithmetic-Trigonometric Optimization Algorithm With Application to Control of Real-Time Pressure Process Plant”, the idea and proposed work is very interesting. I would like to suggest few comments :
- Authors must include a section or subsection to show the core contribution of their proposed work.
- A discussion on the possible limitations of their work may be included to enhance the quality and ethical fairness of their work.
- Add comparative analysis if possible for the proposed work.
- Some typo and grammar mistakes must be removed as well.
The current paper requires moderate English proof read.
Author Response

(The authors gave the same response as above.)
